# Four New Sesquiterpenoids from the Rice Fermentation of *Antrodiella albocinnamomea*

**DOI:** 10.3390/molecules27103344

**Published:** 2022-05-23

**Authors:** Min Guo, Ying-Zhong Liang, Xiu-Ming Cui, Lin-Jiao Shao, Yin-Fei Li, Xiao-Yan Yang

**Affiliations:** 1Faculty of Life Science and Technology, Kunming University of Science and Technology, Kunming 650500, China; guomin@stu.kust.edu.cn (M.G.); lyz@stu.kust.edu.cn (Y.-Z.L.); 20120094@kust.edu.cn (X.-M.C.); shaolinjiao@stu.kust.edu.cn (L.-J.S.); liyinfei@stu.kust.edu.cn (Y.-F.L.); 2Yunnan Key Laboratory of Sustainable Utilization of Panax Notoginseng, Kunming 650500, China

**Keywords:** *Antrodiella albocinnamomea*, sesquiterpene, structure elucidation, sesquiterpenoids, GIAO

## Abstract

Albocimea B-E (**1**–**4**), four new sesquiterpenoids, and four known compounds, steperoxide A (**5**), dankasterone (**6**), 1*H*-indole-3-carboxylic acid (**7**), and (+)-formylanserinone B (**8**), were isolated from the rice fermentation of the fungus *Antrodiella albocinnamomea*. The structures of new compounds were elucidated by comprehensive spectroscopic techniques, the planar structures of new compounds were determined by comprehensive spectroscopic techniques, and their absolute configurations were confirmed via gauge-independent atomic orbital calculations (GIAO), calculation of the electronic circular dichroism (ECD), and optical rotation (OR). These were determined by spectroscopic data analysis.

## 1. Introduction

*Antrodiella albocinnamomea* is a kind of wood-decay higher fungus, which belongs to the family Polyporaceae and is widely distributed in northeast China. The main characteristics are flat fruiting body, two-line hyphae, locked union of reproductive hyphae and saccular body in the fruiting layer [1,2]. Previous research on this fungus led to the isolation of sesquiterpenes and steroids; some of them show the biological activities that are antibacterial, antiprotein tyrosine phosphatase 1B inhibitory, cytotoxic, immunsuppressive, and so on [3,4,5,6,7,8,9]. Sesquiterpenes are the main chemical component in this fungus, including different types of chamigrane, nor-chamigrane, triquinane, gymnomitrane, humulane, a new skeleton, etc. [3,4,6,7,8]. These results inspired us to search for other structurally novel components and bioactive natural products from this higher fungus. We changed the conditions and enlarged the fermentation scale of the fungus, which led to the isolation of four new sesquiterpenoids, albocimea B-E (**1**–**4**), and four known compounds (**5**–**8**) (Figure 1). Detailed spectroscopic analysis and comparison with reported data allowed the determination of four known compounds, steperoxide A (**5**) [10], dankasterone (**6**) [11], 1*H*-indole-3-carboxylic acid (**7**) [12], and (+)-formylanserinone B (**8**) [13].

## 2. Results and Discussion

Compound **1** was obtained as a white amorphous powder. Its molecular formula of C_14_H_20_O_4_ was determined by the HR-ESI-MS at *m*/*z* 275.1255 [M + Na]^+^ (calculated for 275.1254), corresponding to five degrees of unsaturation. The IR absorption bands at 3400, 1706, and 1677 cm^−1^ revealed the presence of hydroxyl, carbonyl, and C=C double bond groups. The ^1^H NMR spectrum (Table 1) revealed resonances for three methyl protons at *δ*_H_ 2.20, 1.14, and 0.98 (each 3H, s), and revealed that those methyl groups were located at quaternary carbons. The ^13^C NMR and DEPT spectra (Table 1) revealed the existence of 14 carbon signals for three methyls, four methylenes, one olefinic methine (*δ*_C_ 121.3), and six quaternary carbons (including two ketone carbonyls at *δ*_C_ 194.7 and 213.1, one oxygenated at *δ*_C_ 91.5, and one olefinic at *δ*_C_ 145.0). These spectroscopic data revealed that **1** should be a dicyclic norsesquiterpenoid. The ^1^H-^1^H COSY spectrum (Figure 2) of **1** established the partial structures of H_1_-4/H_2_-5 and H_2_-9/H_2_-10. The HMBC correlations (Figure 2) from H_3_-12 (*δ*_H_ 1.14) to C-13, C-11, C-10, and C-6, indicated C-12 and C-13 were located on C-11, and C-10 and C-6 were connected by C-11. The HMBC correlations (Figure 2) from H_3_-14 (*δ*_H_ 2.20) to C-7, C-8, and C-6, and H_2_-9 to C-7 indicated C-8 and C-6 were connected by C-7. Beside this, C-5 and C-1 were connected by C-6 from the ^1^H-^1^H COSY correlation of H_1_-4/H_2_-5, and HMBC correlations from H_2_-5 and H_2_-1 to C-6, C-7, and C-11, and so established a 6,6-bicylic skeleton via a C-6 spiro carbon. These data of **1** are closely related to those of antroalbol H [3]. The difference between them is the presence of a double bond group at C-3 and C-4 in compound **1**. This assignment was in accordance with the HMBC correlations (Figure 2) from H-4 (*δ*_H_ 6.03 br s) to C-5, C-6, and C-2, and H_2_-5 to *δ*_C_ 145.0 (C-3). 

In the ROESY spectrum, only weak correlations between H_3_-14 and H_2_-1 can be observed (Appendix A). In order to further determine the relative configurations of **1**, gauge-independent atomic orbital calculations (GIAO), ^13^C NMR calculations, and DP4+ analysis were performed. Compound **1** has two chiral centers, which have four possible configurations. The ^13^C-NMR of four possible structures was calculated at the mPW1PW91/6-31 G (d) level using the GIAO method. Comparison of the ^13^C chemical shifts obtained revealed that the calculated chemical shifts of **1a** and **1c** (Appendix A) were closest to the experimental values. Therefore, the relative configurations of **1** were designated. Finally, the absolute configurations of **1** were assigned as 6*R*,7*S* by comparison of the experimental and calculated ECD (Appendix A). As shown in Appendix A, the calculated curve for 6*R*,7*S* matches well with that of the experimental ECD curve of compound **1.** Thus, the structure of albocimea B (**1**) was determined as depicted.

Compound **2** was obtained as a white amorphous powder, based on a Na^+^ adduct at *m*/*z* 236.1408 [M + Na]^+^ (calculated for 236.1412) via HR-ESI-MS, the molecular formula of **2** was carried out as C_14_H_20_O_3_, suggesting five degrees of unsaturation. The IR absorption band of **2** at 3440 cm^−1^ suggested the presence of a hydroxyl group, while the absorption band at 1712 cm^−1^ suggested the presence of a carbonyl group. The ^1^H NMR spectrum (Table 1) of **2** at *δ*_H_ 1.46, 1.22, and 0.88 (each 3H, s) suggested the existence of three methyl groups attached to quaternary carbons and an aldehyde group (*δ*_H_ 9.67). The ^13^C NMR and DEPT spectra (Table 1) of **2** showed 14 carbon signals attributable to three methyls (*δ*_C_ 26.8, 25.2, and 24.7), four methylenes (*δ*_C_ 37.6, 36.5, 34.3, and 33.8), five quaternary carbons (including one ketone carbonyl at *δ*_C_ 213.4 and one oxygenated carbon at *δ*_C_ 80.0), one aldehyde group (*δ*_C_ 189.5), and a C=C double bond group (*δ*_C_ 150.8, 146.4). The ^1^H-^1^H COSY spectrum (Figure 2) of **2** established the partial structures of H_1_-3/H_2_-4 and H_2_-8/H_2_-9. In the HMBC spectrum (Figure 2), there were correlations from H_2_-1 (*δ*_H_ 2.43) to C-10; H_1_-3 (*δ*_H_ 6.75) to C-1, C-5; H_2_-4 (*δ*_H_ 2.83) to C-2, C-6, C-10; H_2_-8 (*δ*_H_ 2.76) to C-10; H_2_-9 (*δ*_H_ 1.66) to C-11; H_3_-11 (*δ*_H_ 0.88) to C-12; and H_3_-13 (*δ*_H_ 1.46) to C-5, C-7. These data are related to those of spiro[4.5]dec-6-en-1-ol, 2,6,10,10-tetramethyl [14]. Thus, **2** and spiro[4.5]dec-6-en-1-ol, 2,6,10,10-tetramethyl have the same mother nucleus structure and connect different functional groups.

The correlation of H_3_-13 and H_2_-4 is very weak in the ROESY spectrum (Appendix A). In order to further determine the relative configurations of **2**, we adopted the same method as in determining the configurations of compound **1**. Because compound **2** has two chiral centers, it has four possible configurations. ^13^C-NMR calculations of these four possible configurations were carried out at the mPW1PW91/6-31 G (d) level using the GIAO method. By comparison of ^13^C chemical shifts, the relative configurations of **2** can be determined. In order to determine its absolute configuration, the optical rotation (OR) value of configuration (5*S*,6*S*)-**2** was calculated; this value is −9.75. Compared with the experimental OR value of compound **2** (−10.2), the absolute configuration of **2** was finally determined as 5*S*,6*S.* Thus, the structure of albocimea C (**2**) was assigned as depicted.

Compound **3** was obtained as a white amorphous powder with the molecular formula C_16_H_22_O_3_, based on HR-ESI-MS at *m*/*z* 285.1462 [M + Na]^+^ (calculated for 285.1461), which corresponds to six degrees of unsaturation. The IR spectrum absorption band was at 3424 cm^−1^, indicating the presence of a hydroxyl group. The ^1^H NMR spectrum (Table 1) of **3** at *δ*_H_ 3.68, 1.18, 2.29, and 2.19 (each 3H, s) showed the presence of four methyls. The ^13^C NMR and DEPT spectra (Table 1) exhibited 16 carbon signals, including four methyls at *δ*_C_ 51.9, 24.4, 20.6, and 16.4, four methylenes at *δ*_C_ 70.9, 43.0, 42.0, and 35.1, one methyne at *δ*_C_ 124.2, and seven quaternary carbons (including one carboxyl group at *δ*_C_ 172.2). The ^1^H-^1^H COSY spectrum (Figure 2) of **3** did not provide any relevant signals. The HMBC spectrum (Figure 2) showed correlations from H_2_-1 (*δ*_H_ 2.87) to C-9, C-11; H_2_-3 (*δ*_H_ 2.64) to C-8, C-11; H_1_-4 (*δ*_H_ 6.88) to C-3, C-6, C-8; H_2_-10 (*δ*_H_ 3.52) to C-3, C-11; H_3_-12 (*δ*_H_ 2.29) to C-6; H_2_-13 (*δ*_H_ 3.68) to C-5, C-7; H_3_-15 (*δ*_H_ 2.19) to C-6, C-8. These data are closely related to those of compound **M16** in the literature [15]. The difference between **3** and **M16** is the disappearance of the carbonyl group at C-1 in compound **3**. 

The CEs in the CD spectrum are not obvious for **3**; the absolute configuration of this compound was further investigated by comparison of its experimental OR value with those calculated for (2*S*)-**3**. The calculation results show that the calculated OR value of (2*S*)-**3** is +3.01, while the comparative experimental OR value is −2.92. Therefore, it is suggested that the absolute configuration of **3** is opposite to the calculated configuration. Finally, the absolute configuration of **3** was determined as 2*R.* Thus, the structure of albocimea D (**3**) was assigned as depicted.

Compound **4** was isolated as colorless gum. Its molecular formula C_15_H_24_O_3_ was determined on HR-ESI-MS spectrum at *m*/*z* 251.1653 [M − H]^+^ (calculated for 251.1653), corresponding to four degrees of unsaturation. The IR spectrum showed absorption bands for hydroxyl group (3430 cm^−1^) and carbonyl group (1644 cm^−1^). The ^1^H NMR spectrum (Table 1) showed the presence of five methyls at *δ*_H_ 1.71, 1.29, 1.19, 1.07, and 1.05 (each 3H, s). The ^13^C NMR and DEPT spectra (Table 1) showed 15 carbon signals attributable to five methyls (*δ*_C_ 32.1, 31.9, 26.1, 18.0, and 13.2), three methylenes (*δ*_C_ 59.1, 47.4, and 41.2), one methyne (*δ*_C_ 75.5), and six quaternary carbons (including one carbonyl group at *δ*_C_ 207.4). The ^1^H-^1^H COSY spectrum (Figure 2) established the partial structures of H_1_-13/H_3_-14. The HMBC spectrum (Figure 2) displayed correlations from H_2_-1 (*δ*_H_ 2.62) to C-3, C-7, C-9, and C-10; H_2_-3 (*δ*_H_ 1.96) to C-8, C-11; H_2_-4 (*δ*_H_ 1.93) to C-6, C-8; H_3_-10 (*δ*_H_ 1.05) to C-11; H_3_-12 (*δ*_H_ 1.19) to C-6; H_1_-13 (*δ*_H_ 4.04) to C-4; H_3_-14 (*δ*_H_ 1.07) to C-5; H_3_-15 (*δ*_H_ 1.71) to C-6, C-8. These data are related to those of 6,8,8-trimethyl-bicyclo[4,3,0]non-1-en-3-one [16]. Thus, **4** and 6,8,8-trimethyl-bicyclo[4,3,0]non-1-en-3-one have the same mother nucleus structure and connect different functional groups.

The correlations observed in the ROESY spectrum (Appendix A) of **4** were insufficient for determining its relative configuration. Because compound **4** has three chiral centers, it has eight possible configurations. ^13^C-NMR calculations of eight of these possible configurations were carried out at the mPW1PW91/6-31 G (d) level using the GIAO method. The comparison of the ^13^C chemical shifts obtained revealed that the calculated chemical shifts of an enantiomer pair configuration **4a** and **4e** (Appendix A) are the closest to the experimental values. Finally, the absolute configurations of **4** were assigned as 5*R*,9*S*,13*S* by comparison with the experimental and calculated ECD (Appendix A); the calculated curve for 5*R*,9*S*,13*S* matches well with that of the experimental ECD curve of **4**. Thus, compound **4** was established to be albocimea E.

Because the isolated compound materials are limited, only the ones with sufficient amount could be tested for bacteriostatic test. Therefore, compounds **2** and **6** were evaluated for antibacterial activity with the Kirby–Bauer test. The results showed that both had no significant inhibitory activity against *Psecdomonas aeruginosa*, *Staphylococcus aureus*, *Escherichia coli,* and *Monilia albican*.

In conclusion, four previously undescribed sesquiterpenoids (**1**–**4**) and four known compounds (**5**–**8**) were acquired from the rice fermentation of the fungus *A.*
*albocinnamomea*. The structures of these compounds were characterized using spectroscopic data. The antibacterial activity test of compounds **2** and **6** showed that they have no significant antibacterial activity.

## 3. Experimental Section 

### 3.1. General Experimental Procedures

Optical rotations were taken on a JASCO P-1020 polarimeter. IR spectra were obtained on a Bruker Tensor 27 spectrometer with KBr pellets. NMR spectra were measured on a Bruker Avance III 600 MHz spectrometer with TMS as the internal standard. Mass spectra were recorded with an APIQSTAR time-of-flight spectrometer. CD spectra were recorded on an Applied Photophysics spectrometer. Silica gel (200–300 mesh), Sephadex LH-20, and Rp-C_18_ were used for column chromatography (CC). Thin-layer chromatography (TLC) experiments were performed on a silica gel GF_254_ pre-coated plate. Fractions were monitored by TLC, and spots were visualized by spraying with 15% H_2_SO_4_ in ethanol.

### 3.2. Fungal Material and Cultivation Condition

*A. albocinnamomea* was purchased from the China Institute of Microbiology. A voucher specimen (No. Yang20181012) was deposited at the Faculty of Life Science and Technology, Kunming University of Science and Technology. A rice medium was used to ferment the strain. The culture of the strain was divided into two steps. Firstly, the fungal strain was cultured in potato dextrose agar (PDA) medium at 24 °C, and the seed solution was obtained after 7 days of culture. Next, a rice medium was used for large-scale fermentation. The culture medium consisted of rice and water at a ratio of 1:1.4. When preparing the culture medium, 71 g of rice and 100 mL of water were put into 480 mL fermentation bottles. A total of 300 bottles were prepared. They were put into a high-pressure steam sterilization pot and sterilized at 121 °C for 30 min. The seed solution obtained before was divided into small parts, put into the prepared rice medium and incubated at room temperature for 45 days.

### 3.3. Extraction and Isolation

The fungus was cultured for 45 d, cut into small pieces, and then extracted three times with ethyl acetate (60 L × 72 h each time) at room temperature. The ethyl acetate solution was evaporated under vacuum to yield 137.33 g of crude extract. The extract was subjected to CC over silica gel and eluted with CH_2_Cl_2_/MeOH (50:1–10:1) to afford fractions A–F. Fraction C (46.29 g) was subjected to CC over silica gel eluted with CH_2_Cl_2_/MeOH (50:1–10:1) to afford fractions C_1_–C_12_. Subfraction C_4_ (5.32 g) was further purified by silica gel CC (petroleum ether/ethyl acetate, 50:1–0:1), Sephadex LH-20 CC (CH_2_Cl_2_/MeOH, 1:1), silica gel CC (petroleum ether/ethyl acetate, 20:1–0:1), and analytical chromatography to afford **1** (1 mg) and **6** (3.2 mg).

Fraction D (23.24 g) was subjected to CC over silica gel eluted with CH_2_Cl_2_/MeOH (50:1–0:1) to afford fractions D_1_–D_10_. Subfraction D_5_ (3.42 g) was further purified by silica gel CC (petroleum ether/ethyl acetate, 50:1–0:1), Sephadex LH-20 CC (CH_2_Cl_2_/MeOH, 1:1), and analytical chromatography to afford **2** (3.2 mg) and **5** (2.7 mg). D_7_ (237 mg) was further purified by Sephadex LH-20 CC (CH_2_Cl_2_/MeOH, 1:1) to afford **7** (4.7 mg) and **8** (3.6 mg).

Fraction E (40.00 g) was subjected to CC over Rp-18 with H_2_O/MeOH (50–100%) to afford fractions E_1_–E_8_. Subfraction E_6_ (2.5 g) was further purified by silica gel CC (petroleum ether/ethyl acetate, 7:1–0:1), MPLC, and silica gel CC (petroleum ether/ethyl acetate, 4:1) to afford **3** (3.5 mg).

Fraction F (30.00 g) was subjected to column chromatography (CC) over Rp-18 with H_2_O/MeOH (20–100%) to afford fractions F_1_–F_10_. Subfraction F_6_ (4.2 g) was further purified by HPLC using a Sephadex LH-20 CC (CH_2_Cl_2_/MeOH, 1:1) to afford **4** (2.1 mg).

### 3.4. Antibacterial Assays

#### 3.4.1. Bacterial Strain

The strain of *Peseudomonas aeruginosa*, *Staphylococcus aureus*, *Escherichia coli,* and *Monilia albican* were purchased from the Nanjing Bianzhen Biotechnology Co., Ltd (Nanjing, China) and deposited at the Faculty of Life Science and Technology, Kunming University of Science and Technology.

#### 3.4.2. Kirby–Bauer Test

A suspension of the organism to be tested was prepared in a saline solution and measured equal to 0.5 McFarland standard (1 × 10^8^ colony froming units (CFU)/mL). A 0.5 mL amount of bacterial liquid was injected into the nutrient broth culture medium that had been cooled to about 50 °C; this was mixed evenly, poured into the plate (about 20 mL/plate), stood horizontally, and set aside after solidification. The filter paper soaked with the sample was put on the plate with sterile tweezers. Then, the plates were incubated in an incubator at 37 °C for 18 h, and the zones of inhibition were discussed.

## 4. Physical Constants

Albocimea B (**1**): white amorphous powder; ${\left[\alpha \right]}_{D}^{27.7}$ +13.32, (c 0.38, MeOH); IR (KBr) ν_max_ 3401, 2958, 1705, 1676, 1626, 1226 cm^−1^; ^1^H (chloroform-*d*, 600 MHz) and ^13^C NMR (chloroform-*d*, 150 MHz) data, see Table 1; HRESIMS *m*/*z* 275.1255 [M + Na]^+^ (calculated for C_14_H_20_O_4_Na, 275.1254).

Albocimea C (**2**): white amorphous powder; ${\left[\alpha \right]}_{D}^{24}$ −10.2, (c 0.32, MeOH); IR (KBr) ν_max_ 3440, 2957, 2923, 2853, 1703, 1630, 1384, 1272, 1161, 1103, 1063 cm^−1^; ^1^H (chloroform-*d*, 600 MHz) and ^13^C NMR (chloroform-*d*, 150 MHz) data, see Table 1; HRESIMS *m*/*z* 236.1408 [M + Na]^+^ (calculated for C_14_H_20_O_3_Na, 236.1412).

Albocimea D (**3**): colorless oil; ${\left[\alpha \right]}_{D}^{24.8}$ −2.92, (c 0.29, MeOH); IR (KBr) ν_max_ 3436, 2962, 2930, 1733, 1646, 1605, 1436, 1261, 1165, 1033, 803 cm^−1^; ^1^H (chloroform-*d*, 600 MHz) and ^13^C NMR (chloroform-*d*, 150 MHz) data, see Table 1; HRESIMS *m*/*z* 285.1462 [M + Na]^+^ (calculated for C_16_H_22_O_3_Na, 285.1461).

Albocimea E (**4**): colorless gum; ${\left[\alpha \right]}_{D}^{25.0}$ +1.52, (c 0.21, MeOH); IR (KBr) ν_max_ 3424, 2952, 2928, 2868, 1644, 1545, 1509, 1453, 1383, 1284, 1259, 1128, 1104 cm^−1^; ^1^H (methanol-*d*_4_, 600 MHz) and ^13^C NMR (methanol-*d*_4_, 150 MHz) data, see Table 1; HRESIMS *m*/*z* 251.1653 [M − H]^−^ (calculated for C_15_H_23_O_3_, 251.1653).

## Figures and Tables

**Figure 1 molecules-27-03344-f001:**
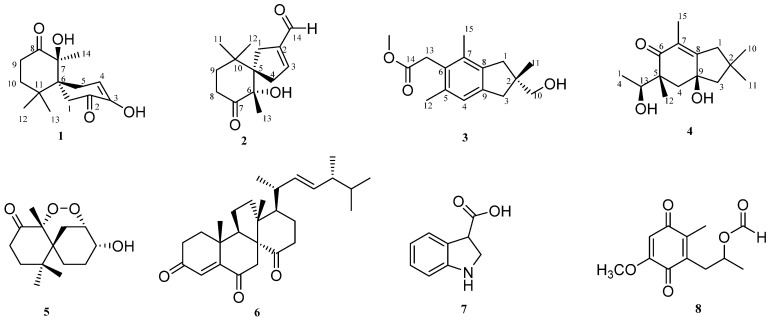
Chemical structure of compounds **1**–**8**.

**Figure 2 molecules-27-03344-f002:**
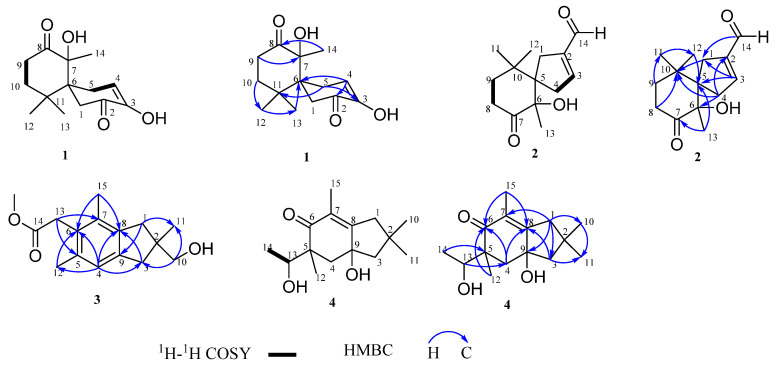
Key ^1^H-^1^H COSY and HMBC correlations of compounds **1–4.**

**Table 1 molecules-27-03344-t001:** ^1^H (600 MHz) and ^13^C NMR (150 MHz) data for compounds **1**–**3** in CDCl_3_ and **4** in CD_3_OD (*δ* in ppm, *J* in Hz).

No.	1		2		3		4	
	*δ* _H_	*δ* _C_	*δ* _H_	*δ* _C_	*δ* _H_	*δ* _C_	*δ* _H_	*δ* _C_
1	2.42, m	32.3	2.43, m	34.3	2.87 t (16.8)	42.0	2.62 br d (18.0) 2.46 d (18.0)	47.4
2		194.7		146.4		44.1		38.9
3		145.0	6.75, br s	150.8	2.64 d (15.9) 2.60 d (15.9)	43.0	1.96 d (18.0) 1.78 d (18.0)	59.1
4	6.03, br s	121.3	2.83, br d (20.3) 2.66, br d (20.3)	37.6	6.88 s	124.2	2.12 d (18.0) 1.93 d (18.0)	41.2
5	2.45, m 1.76, td (13.2, 7.2)	25.2		59.7		135.4		50.6
6		45.5		80.0		129.2		207.4
7		91.5		213.4		133.2		131.5
8		213.1	2.76, td (13.8, 7.0) 2.50, m	33.8		139.5		165.0
9	2.40, m 1.95, m	35.7	1.90, td (13.6, 5.0) 1.66, m	36.5		141.0		78.6
10	1.91, m 1.83, m	38.8		37.0	3.52 s	70.9	1.05 s	32.1
11		55.5	0.88, s	26.8	1.18 s	24.4	1.29 s	31.9
12	1.14, s	24.3	1.22, s	24.7	2.29 s	20.6	1.19 s	26.1
13	0.98, s	24.4	1.46, s	25.2	3.68 m	35.1	4.04 dd (13.2, 6.6)	75.5
14	2.20, s	26.8	9.67, s	189.5		172.2	1.07 d (6.0)	18.0
15					2.19 s	16.4	1.71 s	13.2
-OCH_3_					3.68 s	51.9		

## Data Availability

Not applicable.

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
