# Peer review of "Four New Sesquiterpenoids from the Rice Fermentation of Antrodiella albocinnamomea"

_molecules, 2022, doi:10.3390/molecules27103344_

Round 1
Reviewer 1 Report
The paper entitled "Four New Sesquiterpenoids from the Rice Fermentation of Antrodiella albocinnamomea" describes the isolation and the structural elucidation fo 4 new compounds obtained from Antrodiella albocinnamomea. The paper il well written and the data are well presented. There are some typos that do not preclude the reading of the manuscript. In the sump. info, I suggest adding the structures of the compounds to the NMR spectra to facilitate reading. My overall suggestion is to accept the paper with minor revisions

Round 2
Reviewer 2 Report
- Since the stereochemistry of the compounds was determined by the same methods (13C-NMR calculations, ECD or OR calculation), authors should first introduce the planar structural determination first and then introduce stereochemistry determination by 13C NMR, ECD calculation and OR calculation. This will make the whole structural elucidation part smoother.
- Authors should pay attention to the details. For instance, the term HR-EI-MS is different from HR-ESI-MS. The authors apparently used ESI, not EI. No Na adduct could be observed in EI-MS. Please double check the whole manuscript and revise them all.
- In page 3, lines 41-43. The authors should rethink the way they present. As I mentioned before, there is no way one can determine the absolute configuration(s) of a compound by 13C NMR calculation, no matter how many chiral centers it has. Please delete this sentence:” Because the absorption band of ECD spectrum of compound 3 is small, ECD calcu-41 lation cannot be carried out to determine the configuration, and there is only one chiral 42 center, so it is impossible to determine the configuration by 13C-NMR calculation through 43 GIAO.” Also, what authors wanted to say is the CEs in the CD spectrum is not obvious not the absorption band of ECD spectrum of compound 3 is small.
- The authors should attach the chiral analysis of 3 and 4 in supporting information to verify the isolates are optically pure.
- Page 4, lines 22-24. For the known compounds, they could be readily determined. So I suggest to incorporate this part into introduction accordingly.
- The results showed that both had no significant inhibitory activity against Psecdomonas aeruginosa, Staphylococcus aureus, Escherichia coli and Monilia albican. Where is the data? Please attached the pictures in si. Why only 2 and 6 were tested? Authors should mention due to the limited material of all the isolates, only the ones with sufficient amount were tested.
- In figure 2, the carbon numbers and HMBC correlations are overlapped. Please revise.
